# Research Capacity Training on Environmental Health and Noncommunicable Diseases in the Country of Georgia: Challenges and Lessons Learned during the COVID-19 Pandemic

**DOI:** 10.3390/ijerph19138154

**Published:** 2022-07-02

**Authors:** Carla J. Berg, Lela Sturua, Carmen J. Marsit, Levan Baramidze, Nino Kiladze, William Michael Caudle

**Affiliations:** 1Department of Prevention and Community Health, Milken Institute School of Public Health, George Washington University, Washington, DC 20052, USA; 2Non-Communicable Diseases Department, National Center for Disease Control and Public Health, 0198 Tbilisi, Georgia; lela.sturua@ncdc.ge; 3Gangarosa Department of Environmental Health, Rollins School of Public Health, Emory University, Atlanta, GA 30322, USA; carmen.j.marsit@emory.edu (C.J.M.); william.m.caudle@emory.edu (W.M.C.); 4International School of Public Health, Tbilisi State Medical University, 0177 Tbilisi, Georgia; l.baramidze@tsmu.edu (L.B.); nkiladze@tsmu.edu (N.K.)

**Keywords:** global health training, global health, COVID-19, mentorship, environmental health, noncommunicable diseases, low- and middle-income countries

## Abstract

COVID-19 presented challenges for global health research training programs. The Clean Air Research and Education (CARE) program, which aims to enhance research capacity related to noncommunicable diseases and environmental health in the country of Georgia, was launched in 2020—as the COVID-19 pandemic began. At its foundation is mentorship and mentored research, alongside formal didactic training, informal training/meetings, and other supports. Current analyses examined CARE’s initial 1.5 years (e.g., program benefits, mentorship relationships) using data from an evaluation survey among trainees and faculty in January 2022. Trainees (100% response rate: *n* = 12/12; 4 MPH, 8 PhD) and faculty (86.7% response rate: *n* = 13/15; 7 Georgia-based, 6 United States-based) rated factors related to mentor-mentee relationships highly, particularly mutual consideration of each other’s thoughts, opinions, and perspectives; one major challenge was completing goals planned. Trainees and faculty identified several growth experiences and program benefits (e.g., skills development, expanding professional network) but also identified challenges (e.g., meeting program demands, communication gaps, unclear expectations)—exacerbated by the pandemic. Findings underscore the importance of strong mentorship relationships and that the pandemic negatively impacted communication and clarity of expectations. Given the likely ongoing impact of the pandemic on such programs, program leaders must identify ways to address these challenges.

## 1. Introduction

Training programs in low- and middle-income countries (LMICs) are a critical component of meeting the goal of building global research capacity. A number of research training programs have highlighted the importance of training for health researchers. One of the largest institutions funding this type of education and training is the United States (US) National Institutes of Health (NIH) Fogarty International Center (FIC), dedicated to supporting and facilitating partnerships between health research institutions in the US and LMICs around the globe, and training scientists to address their local health needs on a global scale [1].

Such global health training programs may involve short- (3-month) to long-term (>6-month) training [2,3], with long-term training potentially involving formal graduate education such as master’s, doctoral, and postdoctoral degree programs related to the training areas needed and health concerns within trainees’ home countries [3,4]. Moreover, these training programs may take multiple approaches, such as bringing trainees to grantee institutions in developed countries to participate in training (e.g., in-person coursework, skills-based workshops) and work directly with researchers and mentors, travel by mentors in the grantee institutions to partner institutions in LMICs to facilitate such training, online or distance based courses and training, or some blend of the approaches [4].

FIC D43 training programs are funded through peer-reviewed grants and designed to be collaborative, long-term, and flexible to meet the research priorities of both the US and foreign institutions [2,5]. This paper focuses on the experiences of an FIC D43, co-funded by the National Institute of Environmental Health Sciences (NIEHS), in the Republic of Georgia (GE). Launched in 2020, the Clean Air Research and Education (CARE) program is a collaboration between Emory University, the Georgia National Centers for Disease Control and Public Health (NCDC), Tbilisi State Medical University (TSMU), and George Washington University. CARE has the long-term goal of enhancing the capacity of GE to conduct research related to noncommunicable diseases (NCDs) and environmental health (EH) to inform related policy and practice ultimately [6]. In context, 7 million deaths worldwide are attributable to the joint effects of indoor and ambient air pollution annually, with ~94% occurring in LMICs [7]. While 51% of cities in high-income countries with ≥100,000 residents meet WHO air quality guidelines, only 3% of such cities in LMICs meet them [8]. In GE, adverse environmental exposures cause 21% of disease burden and 25% of deaths [9], including 30% of disease burden and 14% of deaths among children [9]. According to 2016 WHO data, GE’s mortality index attributed to ambient and indoor air pollution was 204.9, the 3rd highest in the world [10]. Accordingly, GE’s 2017–2021 National Environment and Health Action Plan (NEHAP-2), which is conceptually and strategically linked with the United Nations’ 2030 Sustainable Development Goals and Health 2020, highlights that addressing air pollution is among the most prominent public health priorities.

Despite the importance of addressing EH and NCDs in GE, there is limited in-country capacity to conduct research regarding the impact of such environmental hazards on health. The CARE program aims to address this gap by implementing a research capacity-building program via training and applied research training opportunities for MPH and PhD trainees to address NCDs and EH. A core foundation of the program is strong mentorship via mentor pairs across GE and the US, given the importance of effective mentorship in the development, success, and retention of trainees and early career investigators in academic research settings [11,12,13,14,15,16,17].

The timeline for the CARE program, unfortunately, coincided with the beginning of the COVID-19 pandemic. Indeed, since January 2020, the pandemic has caused the “lockdown” of billions of people and millions of deaths globally [18]. The pandemic has created adverse economic and social consequences that have directly and indirectly impacted global health activities, including global research training programs in LMICs. For example, training activities previously implemented in-country (e.g., on-site components requiring travel by US faculty and mentors) or in the US (e.g., US-based research experiences) that historically were critical in team-building and sociocultural adoption have been hindered (e.g., postponed, canceled, replaced by virtual communications/meetings). While such changes have had some positive impacts (e.g., lower program costs, opportunities to include a wider range of faculty for training activities), other impacts have been less positive, such as lost opportunities to build the mentor-mentee relationship and to expose mentees to other cultures and ideas through international travel [18].

This paper provides data regarding the evaluation of the first 1.5 years of the CARE program—coinciding with the challenges surrounding the COVID-19 pandemic, integrating perspectives of both trainees and faculty. Specifically, we examined themes regarding key opportunities for training and desired resources and supports, as well as experiences with the mentor-mentee relationship, including challenges faced and suggestions for future program implementation.

## 2. Materials and Methods

### 2.1. Training Program Description

The CARE Research Training Program, founded in September 2019 and accepted its first cohort of MPH and PhD trainees in September 2020, enhances research capacity via formal didactic training, informal training via meetings/workshops, mentorship and mentored research, and other instrumental supports to execute trainee research. Figure 1 provides an overview of activities since the program’s launch.

*Formal Training*. CARE was initially designed to integrate various training formats (e.g., distance learning, in-person intensive short courses); however, due to the pandemic, all training has been virtual to date. Two core courses have been offered each year to the incoming cohort. A core EH course is held virtually every spring and taught by a US-based faculty member. This course was offered as a 2-week course in March 2021 but then revised as a 4-week course offered in March–April 2022, expanded in the timeline in response to trainee feedback on the intensity of the 2-week course. Additionally, a Research Methods course, covering both quantitative and qualitative methods, is held in a hybrid format, taught by an expert team of GE-based faculty, and funded by the CARE program across several months (e.g., April–September 2021). Additional high-priority training are determined each year via assessments among trainees and faculty.

*Semi-annual CARE Meetings*. Semi-annual meetings are held in June and October/November each year, with the fall meeting serving as an orientation to the new cohort of trainees. These meetings entail comprehensive training in responsible conduct of research, trainee presentations of the proposed and ongoing research, keynote lectures (e.g., public health communication), and workshops on special topics (e.g., communication skills, mentorship). To date, these meetings have been held in a hybrid format, with the majority of GE-based faculty and trainees participating in person and US-based faculty participating virtually.

*Monthly CARE Fellow Meetings*. All trainees participate in regularly occurring virtual CARE fellow meetings, which were held monthly from Spring 2021 and then twice per month beginning in January 2022. These meetings have included special topics (e.g., “imposter syndrome,” mentorship, communication, and writing skills), trainee updates on their research projects, and research presentations by GE- and US-based faculty.

*Mentorship & Mentored Research Activities*. Mentorship is provided via pairs of mentors from each country. Because of the importance of mentorship in the program, several activities aim to enhance related experiences and skills among trainees and faculty. For example, during the October/November CARE meeting that serves as an orientation to the new cohort of trainees, a special session, driven by the robust literature on successful mentor-mentee relationships [11,12,13,14,15,16,17], is devoted to underscoring the roles of mentors and mentees, the goals of the mentor-mentee relationship, and how to work together as mentors and mentees successfully. This is then reinforced throughout the year via the monthly CARE fellow meetings and via quarterly meetings among faculty serving as mentors. Mentored research activities are designed to either coincide with ongoing faculty research focused on NCDs and EH or allow trainees to execute their own study, and trainees are asked to meet with their faculty mentors as a group every month to discuss their progress.

*Other Instrumental Supports to Conduct Research.* Trainees are provided additional assistance related to English language and writing, statistical consultation, and other supports on a case-by-case basis. They are also provided access to funds to support the execution of their thesis and dissertation research, and as well as opportunities to participate in scientific conferences (albeit limited due to the pandemic); for example, trainees participated in the virtual GE-based Health & Ecology Conference in October 2021.

### 2.2. Measures

In January 2022, program leadership administered an online survey to evaluate the initial 1.5 years of the program, including its overall activities, training, and experiences with mentorship among both trainees and faculty. This survey was administered to trainees, faculty, and mentors in the program to assess the individual experiences of members of each of these groups and gain insight to improve across all stakeholders. The survey was constructed by adapting existing published measures (noted below) and then piloted among members of our team to ensure clarity and comprehension.

*Participant Characteristics.* The evaluation assessed trainee and faculty gender, years in the program (since 2020 or 2021), MPH versus PhD track among trainees, or GE- or US-based among faculty.

*Program Benefits and Resource Utilization.* To assess necessary and desired training program resources, trainees were asked: (1) “How has/will your participation in the Fogarty D43 CARE program help you achieve your goals?” (2) “What benefits of the CARE program do you anticipate using (short-term US-based training in public health; access to funds to support your dissertation/thesis work; access to funds to support attending/presenting at national/regional/international scientific meetings; other; none of the above)?” and (3) “Please indicate what trainings, workshops and other professional development activities you think will be beneficial to you and your professional growth (e.g., specific classes/topics, meetings, visits to the US or elsewhere, etc.).” Faculty were asked, “What ideas do you have about training for this program that you think would be beneficial to trainees?”

*Experiences with Mentor/Mentee Relationship.* To evaluate mentor/mentee relationships and processes among both trainees and faculty, parallel items were used. Participants [trainees/faculty] were instructed, “This section asks about your experience with your [mentors/mentees] to date. Who are your [mentors/mentees] in the program? (Check all that apply).” Trainees were asked to evaluate their mentorship team overall; faculty were asked to evaluate each individual mentee. They were then asked: “Please indicate your responses to the following questions regarding your [mentor/mentee(s)]: (1) Are you happy with the frequency of meetings? (2) Did/do you find the meetings productive? (3) Did [your mentors work with you/you help your mentee] to identify tangible steps to meet goals and objectives? (4) Did [your mentors connect you/you connect your mentee] with other professionals who could “fill in the gaps” in areas where mentors might be less skilled? (5) Did you and your [mentors/mentee] complete the goals planned? (6) Were your [mentors/mentee] responsive to your emails and other forms of communication? (7) Did/do [your mentors consider your perspective/you consider your mentee’s perspective] and respect mentee goals and objectives? (8) Did/do [your mentors consider your/your mentee consider your] advice and accept encouragement with respect to mentee goals and objectives? (9) Did/do [your mentors/you] solicit [your/your mentee’s] thoughts and opinions when making suggestions or recommendations? (10) Are you satisfied with your relationships with your [mentors/mentee]?” Response options included: not at all; sometimes/somewhat; the majority of the time; almost always/always; don’t know; and not applicable [19,20]. The scale demonstrated high internal consistency (Cronbach’s α = 0.79 for trainees; Cronbach’s α = 0.96 for faculty). Factor analysis indicated two factors: (1) communication and relationship quality (accounting for 73% of the variance); and (2) instrumental support (accounting for 12%).

Qualitative data were also obtained via open-ended questions. Participants [trainees/faculty] were asked: (1) “What have you learned about yourself as a result of being a [mentee/mentor] in this program? If you are new to the program, what do you hope to learn about yourself as a result of [receiving mentorship/being a mentor]?” (2) “What challenges have you found or do you anticipate regarding the mentor/mentee relationship in this program specifically?” and (3) “What suggestions do you have for improving the mentor/mentee experiences in this program? If you are new to the program, what do you hope to have as part of the mentor/mentee experience?”

### 2.3. Data Analysis

Descriptive analyses were conducted using SPSS version 26.0 (IBM, Armonk, New York, NY, USA) to characterize the trainee and faculty participants and their responses to the closed-ended items regarding experiences with the mentor/mentee relationships. Responses to open-ended questions were qualitatively analyzed and presented.

## 3. Results

### 3.1. Participant Characteristics

Response rates were 100% among trainees (*n* = 12/12) and 86.6% among faculty (*n* = 13/15). Table 1 provides data regarding participant gender (9 male, 16 female), time in the program (13 since 2020, 12 since 2021), program track among trainees (4 MPH, 8 PhD), and country among faculty (7 GE-based, 6 US-based).

### 3.2. Program Benefits and Resource Utilization

Trainees provided various perspectives regarding how their participation in the program will facilitate the achievement of their goals (not shown in tables). Themes included: (1) advancing knowledge and skills needed for success as a research scientist; (2) support professional and career development; (3) opportunity to conduct independent research on the topic of interest; (4) opportunity to collaborate and work on a research topic with internationally-recognized expert mentors; (5) boost networking abilities to establish the foundation for career and research; (6) training to transform theory and evidence into practice.

Trainees indicated great interest in: (1) short-term US-based training in public health (*n* = 9/12); (2) access to funds to support attending/presenting at national/regional/international scientific meetings (*n* = 5/12), and (3) access to funds to support dissertation/thesis work (*n* = 4/12).

Table 2 provides an overview of themes of participants’ responses regarding key opportunities for training in the CARE program, identifying several similarities across trainees and faculty regarding untapped opportunities to date. Similar responses underscoring the merit of training already underway included: advanced statistics, data analyses; research methodology; scientific/grant writing, public speaking course, English language; epidemiology; and special topics; faculty also underscored the need to provide training related to professional development (partially addressed through fellow club meetings). Other suggestions from trainees (geographic information system [GIS]) and faculty (i.e., public health policy, evidence-based decision-making) have not yet implemented but are being incorporated into the program’s training plans.

### 3.3. Experiences with Mentor/Mentee Relationships

With regard to experiences with the mentor/mentee relationships, evaluations were generally positive (Table 1). The mode response for almost all items for trainees and faculty being 4 out of 4 (all for trainees, all but one for faculty: i.e., “Did you and your mentee complete the goals planned?”), and with over half of items having a median response of 4 (all but one for trainees: i.e., “Did/do you and your mentors complete the goals planned?”; and half of the faculty responses had a median of 3). Among both trainees and faculty, the highest ratings were for the items assessing “Solicits [mentor’s/mentee’s] thoughts and opinions when making suggestions or recommendations” and “Consider [mentor’s/mentee’s] perspective and respect mentee goals and objectives.” The lowest rated item for both trainees and faculty was “You and your [mentors/mentee] completed goals planned,” with 2 exceptions among faculty (i.e., “Happy with the frequency of meetings,” “Mentor connected mentee with other professionals who could “fill in the gaps” in areas where they might be less skilled”).

Table 2 provides an overview of themes of participants’ responses to the items assessing what they learned about themselves from participating as a mentee/mentor in the CARE program, which represents clear distinctions in the experiences of trainees vs. faculty. Trainees indicated that they learned the importance/benefits of teamwork, mentor support, attention to detail and follow-through, and working with potential future colleagues, as well as how to receive constructive feedback and the time demands of the training. They also indicated that they hoped to develop skills in forming their ideas, critical thinking, and effectively communicating.

Faculty indicated that they had learned that expectations, ways of communication, and academic structures differ between GE and US—which also included differences in the nature of mentor-mentee relationships. They indicated that they had expanded their mentoring skills by mentoring students with different needs but needed more support navigating this process. Additionally, they learned the importance of opportunities for mentor/mentee professional networking via the CARE program. They also indicated that they hoped to develop an expanded network of colleagues and acquire greater skill in mentoring students, particularly from a distance, and advanced trainees.

With regard to challenges in the program to date (Table 2), the themes across trainees and faculty were similar, including meeting the demands of the program (e.g., frequency of meetings, time for mentoring opportunities); communication, gaps in communication, misunderstandings as a result of convoluted communication; and lack of clarity regarding expectations. Additionally, trainees underscored the challenges related to having 2–3 mentors who vary in expectations, contributions, etc., and executing the work and writing related to the thesis/dissertation. Faculty included additional challenges as well, including limited progress/commitment from select trainees and the need for in-person meetings of all team members (i.e., the impact of COVID-19 on the in-person engagement of US-based mentors).

Regarding program suggestions, trainees and faculty also outlined some similar themes (Table 2), including more communication, more in-person meetings, and greater clarity regarding timelines and expectations related to the thesis/dissertation. Trainees also mentioned the need for all mentors to have a shared understanding of expectations, more autonomy in the mentorship and training process, and more opportunities for mentors to share practical advice and support. Faculty provided additional suggestions, including more meetings among the mentors, as well as written materials, to facilitate a shared understanding of expectations; greater clarity regarding the roles of GE- vs. US-based mentors and any distinctions; the need to dismantle hierarchical barriers between trainees and faculty to facilitate collegial relationships between mentors-mentees; and more careful selection of trainees admitted to the program and anticipating challenges to be timely in responding to them.

## 4. Discussion

The COVID-19 pandemic has transformed global health training by making the exploration and use of virtual opportunities for training and mentorship a necessity, as shown by our examples and those of others [18]. Despite the impact of the pandemic on the planned in-person activities and other experiences requiring travel (e.g., scientific meetings, in-person semi-annual trainee/faculty meetings), both trainees and faculty rated factors related to the mentor-mentee relationships highly. The dimensions rated highest among both trainees and faculty were with regard to mutual consideration of each other’s thoughts and opinions when making suggestions or recommendations” and “perspective and respect” for mentee goals and objectives”. Among the lowest-rated items for both trainees and faculty related to completing goals planned. Despite broad acceptance that faculty can benefit from mentoring training [12,14], there are few formal mentorship training programs providing best practices for mentoring in LMIC settings [11,13]. Current findings indicate that our integration of training regarding the mentor-mentee relationship for both trainees and faculty may provide the foundation for a shared understanding of one another’s roles—and thus facilitate more successful interactions in the short term—and hopefully in the long term [13].

Our findings indicated that trainees valued the opportunities provided by the program (i.e., advanced skills, professional development, professional network), also underscored by trainee outcomes in prior training programs [5]. Trainees also specifically identified high-priority training (e.g., advanced methods/analyses, specialized topics) and key program resources they intended to use, 2 of which have been hindered by the pandemic (i.e., short-term US-based training, support to attend scientific meetings). They also reported gleaning benefits from the program to date (e.g., importance/benefits of teamwork, mentor support, attention to detail and follow-through, and working with potential future colleagues).

However, both trainees and faculty identified challenges related to meeting program demands, communication gaps, and limited clarity of expectations, which were further underscored by trainee and faculty suggestions for the program in the future (e.g., more communication, more in-person meetings, greater clarity regarding timelines and expectations). These challenges have also been discussed in the literature regarding such training programs during the pandemic. While virtual approaches can be effective at a lower cost, they may also undermine the effectiveness of such training programs on trainee professional development due to weakened or lost in-person interactions, team building, sociocultural adoption and understanding, and interpersonal relationships [16]. Given the altered formats and timelines of training and the possibility of long-term adoption of these changes [4], future programs may attract different types of learners with different and varied motivations, expectations, and outcomes [21]. The development of new “hybrid” models using proven virtual components but with built-in, in-person activities better suited to discussion, idea generation, and team and relationship building may provide the best approaches for global health training and education in the post-COVID-19 era [4].

Limitations to this study include the self-reported nature of the items, as well as the potential for bias reporting among both trainees and faculty. In addition, the qualitative data was limited, as it was assessed using open-ended questions with open fields to which participants responded rather than in-depth interviews, which would have provided greater depth to our survey findings.

## 5. Conclusions

Trainees perceived great benefit of program participation and mentorship. Moreover, integrating training regarding the mentor-mentee relationship for both trainees and faculty may provide the foundation for more successful interactions. However, trainees and faculty perceived communication and lack of clear expectations to be challenges, both of which may be compounded by the loss of in-person meetings and the ability to build strong relationships and shared understandings of expectations and roles, particularly relevant in collaborations across different cultural norms. Thus, it is essential to identify innovative ways to make virtual meetings and platforms more conducive to communication (e.g., project collaboration and planning tools) and relationship building (e.g., sharing personal information/photos, connecting via ice-breaker questions). This is particularly crucial given that many global health training programs were already shifting online, even before the COVID-19 pandemic. The likely long-lasting impact of the pandemic has solidified the need to ensure that global health research training programs capitalize on both in-person and virtual opportunities to enhance mentee-mentor relationships, facilitate better communication, and foster the achievement of trainee goals.

## Figures and Tables

**Figure 1 ijerph-19-08154-f001:**
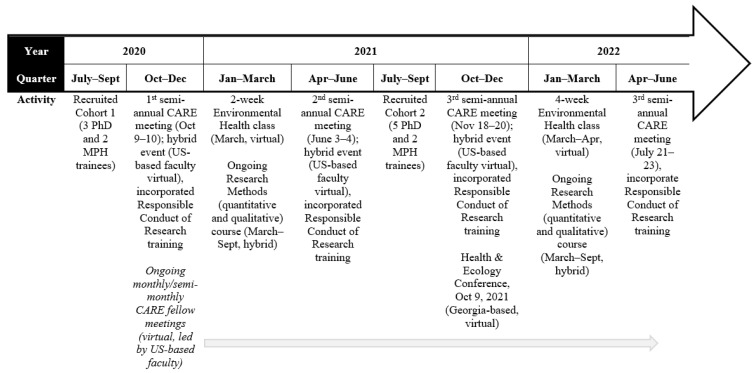
CARE Training Program activities.

**Table 1 ijerph-19-08154-t001:** CARE Training Program participant characteristics and mentor/mentee evaluations.

Variable	Trainees *n* = 12	Faculty *n* = 13
** *Participant Characteristics* **		
Gender		
Male	3	6
Female	9	7
Years in program		
Since 2020	5	8
Since 2021	7	5
Trainee track		
MPH	4	--
PhD	8	--
Country		
Georgia	--	7
US	--	6
** *Mentor/Mentee Evaluations* **		
*Communication and relationship quality:*		
Solicits [mentor’s/mentee’s] thoughts and opinions when making suggestions or recommendations	3.91 (0.30) ^a^	3.41 (0.78)
Trainee considers mentor’s advice and accepts their encouragement with respect to goals and objectives	3.83 (0.39) ^a^	3.36 (0.83)
[Mentors/mentee] responsive to emails and other forms of communication	3.83 (0.39)	3.30 (0.84)
Satisfaction with mentor/mentee relationships	3.83 (0.58)	3.10 (1.00)
Happy with the frequency of meetings	3.73 (0.65) ^a^	2.90 (0.96)
Find the meetings productive	3.58 (0.67)	3.20 (0.96)
*Instrumental support:*		
Consider [mentor’s/mentee’s] perspective and respect mentee goals and objectives	3.91 (0.30)	3.43 (0.86)
Mentor connected mentee with other professionals who could “fill in the gaps” in areas where they might be less skilled	3.63 (0.74) ^b^	2.93 (1.10)
Worked together to identify tangible steps to meet mentee goals and objectives	3.33 (0.89)	3.31 (0.89)
You and your [mentors/mentee] completed goals planned	3.33 (0.78)	2.97 (0.96)

Notes: Missing data: ^a^ 1 Don’t know response. ^b^ 1 Don’t know and 3 Not applicable responses.

**Table 2 ijerph-19-08154-t002:** Themes from questions assessing key opportunities for training, personal/professional development in the program, challenges faced, and suggestions for improving the program.

Program Benefits and Resource Utilization	
Trainees	Faculty
** *What training, workshops, and other professional development activities you think will be beneficial to you and your professional growth?* **	** *What ideas do you have about training for this program that you think would be beneficial to trainees?* **
− Advanced statistics, data analyses − Research methodology − Scientific/grant writing, public speaking course, English language − Epidemiology − Special topics linked to common research themes (e.g., tobacco control, air pollution, urban health, health impact analysis, built environment) − Geographic information system (GIS)	− Advanced statistics, data analyses − Research methodology − Scientific/grant writing, public speaking course, English language − Epidemiology − Special topics (e.g., noncommunicable disease prevention, public health laboratory issues) − Public health policy, evidence-based decision-making − Professional development (e.g., time management, mental health, general navigation of a PhD/graduate training)
**Experiences with Mentor/Mentee Relationship**	
** *What have you learned about yourself from being a [mentee/mentor] in this program? If new, what do you hope to learn?* **	
**Trainees**	**Faculty**
*Have learned:* − Benefits of teamwork − Importance of mentor support − Value my attention to detail and follow-through − Importance of working with potential future colleagues − How to receive constructive feedback − Need to devote more time to the process than initially anticipated − *Hope to develop:* − How to ask questions, gain confidence to share ideas, and develop strong communication skills. − Manage issues interfering with the formation and transmission of thoughts − Critical thinking, focus on priority issues, and writing skills	*Have learned:* − Expectations, ways of communication, and academic structures differ between Georgia and US, and the need to adapt accordingly − Culture of close mentor-mentee relationships − Expanded mentoring skills by mentoring students with different needs − Importance of giving back my knowledge and experience − How to be responsive and supportive to all mentees − Importance of opportunities for mentor-mentee professional networking − *Hope to develop:* − Expand networking to have close contact with colleagues from the US − Acquire new skills in mentoring and supporting mentees, particularly from a distance, and particularly advanced doctoral trainees
** *What challenges have you found or do you anticipate regarding the mentor/mentee relationship in this program specifically?* **	
**Trainees**	**Faculty**
− Meeting the demands of the program (e.g., frequency of meetings, time for mentoring opportunities) − Communication, gaps in communication, misunderstandings as a result of convoluted communication − Lack of clarity regarding expectations − Having 2–3 mentors who vary in expectations, contributions, etc. − Executing the work and writing related to the thesis/dissertation	− Meeting the demands of the program (e.g., frequency of meetings, time for mentoring opportunities) − Communication, gaps in communication, misunderstandings as a result of convoluted communication − Lack of clarity regarding expectations − Limited progress/commitment from select trainees − The need for in-person meetings of all team members (i.e., the impact of COVID-19 on the in-person engagement of US-based mentors)
** *What suggestions do you have for improving mentor/mentee experiences?* ** * **If new, what do you hope to have as part of the mentor/mentee experience?** *	
**Trainees**	**Faculty**
− More communication − More in-person meetings − Clarify timelines and expectations regarding thesis/dissertation − Need for all mentors to have a shared understanding of expectations − More autonomy in the mentorship and training process − More opportunities for mentors to share practical advice and support	− More communication − More in-person meetings − Clarify timelines and expectations regarding thesis/dissertation − More meetings among the mentors, as well as written materials, to facilitate a shared understanding of expectations − Clarify roles of in-country vs. US-based mentors, and any distinctions − Dismantle hierarchical barriers to facilitate collegial peer relationships between mentors-mentees − More careful selection of trainees admitted to the program and anticipating challenges to be timely in responding to them

## Data Availability

Data not publicly available (available upon request).

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
