# Peer review of "Research Capacity Training on Environmental Health and Noncommunicable Diseases in the Country of Georgia: Challenges and Lessons Learned during the COVID-19 Pandemic"

_ijerph, 2022, doi:10.3390/ijerph19138154_

Round 1

Reviewer 1 Report

This manuscript presents a study of Research capacity training on environmental health and non-communicable diseases in the country of Georgia: challenges and lessons learned during the COVID-19 pandemic. The topic is relevant and the study design and the study procedure are very clear. The article has a clear language.    However, I have several suggestions:

Abstract

The authors have used some abbreviations that need to be spelled out, such as GE, US.  
The objectives and conclusion are not clear, in the abstract.

Material and methods

Line 153-“… an online survey to evaluate…” Has the questionnaire been validated? In studies that use questionnaires, validation of the questionnaires is important because a valid questionnaire helps to collect data of better quality and with high comparability, which reduces the effort and increases the credibility of the data

Results

Table 1-the presentation of results regarding the mentor/mentee evaluations part is very confusing. Perhaps, the authors should put some of the information they have in the notes, in the table. They should avoid using * or ** to indicate results

Table 2 Please spell out “NCD”   for de readers to understand the name, without needing to read the rest of the article

Author Response

We would like to thank the reviewers for their thoughtful feedback. Below we respond to each point. We appreciate the opportunity to enhance the quality of this manuscript by doing so.

Reviewer 1

  1. This manuscript presents a study of Research capacity training on environmental health and non-communicable diseases in the country of Georgia: challenges and lessons learned during the COVID-19 pandemic. The topic is relevant and the study design and the study procedure are very clear. The article has a clear language.  However, I have several suggestions.

Response: Thank you for the favorable comments and for the very helpful suggestions below.

  1. Abstract: The authors have used some abbreviations that need to be spelled out, such as GE, US.  The objectives and conclusion are not clear, in the abstract.

Response: We have used the spelled out version of GE (Georgia) and US (United States) and have clarified out objectives so that the conclusions are also clarified. We have written: “Current analyses examined CARE’s initial 1.5 years (e.g., program benefits, mentorship relationships) using data from an evaluation survey among trainees and faculty in January 2022…. Findings underscore the importance of strong mentorship relationships, and that the pandemic negatively impacted communication and clarity of expectations. Given the likely ongoing impact of the pandemic on such programs, program leaders must identify ways to address these challenges.”

  1. Material and methods: Line 153-“…an online survey to evaluate…” Has the questionnaire been validated? In studies that use questionnaires, validation of the questionnaires is important because a valid questionnaire helps to collect data of better quality and with high comparability, which reduces the effort and increases the credibility of the data

Response: Thank you for noting this important consideration. As noted in the measures section, we derived the closed-ended measures assessing mentoring relationships from published measures, and we piloted our open-ended questions among our study team to ensure clarity and comprehension. To be more explicit about this process, we have included: “The survey was constructed by adapting existing published measures (noted below), and then piloted among members of our team to ensure clarity and comprehension.”

  1. Results: Table 1-the presentation of results regarding the mentor/mentee evaluations part is very confusing. Perhaps, the authors should put some of the information they have in the notes, in the table. They should avoid using * or ** to indicate results

Response: Thank you for noting this. We have alternatively included those notes as appropriate in text. Specifically, internal consistency and factor structure are now indicated in the measures section: “The scale demonstrated high internal consistency (Cronbach’s α=.79 for trainees; Cronbach’s α=.96 for faculty). Factor analysis indicated 2 factors: 1) communication and relationship quality (accounting for 73% of the variance); and 2) instrumental support (accounting for 12%).” Data regarding mode and median were integrated into the text of the results section. The only notes remaining are in relation to missing responses.

  1. Table 2 Please spell out “NCD”  for de readers to understand the name, without needing to read the rest of the article

Response: We have done so.

Reviewer 2 Report

The manuscript entitled "Research capacity training on environmental health and noncommunicable diseases in the country of Georgia: challenges and lessons learned during the COVID-19 pandemic" by Carla Berg et al. presents data about an international global health research training and its challenges and opportunities, especially in the background of COVI-19 pandemic. The manuscript is well written and the results are represented in a sufficient and clear way.

In summary the manuscript can be recommended for publication, However, it would be desirable if the authors could add a paragraph describing how they would address the challenges mentioned here in further programs.

Author Response

We would like to thank the reviewers for their thoughtful feedback. Below we respond to each point. We appreciate the opportunity to enhance the quality of this manuscript by doing so.

Reviewer 2

  1. The manuscript entitled "Research capacity training on environmental health and noncommunicable diseases in the country of Georgia: challenges and lessons learned during the COVID-19 pandemic" by Carla Berg et al. presents data about an international global health research training and its challenges and opportunities, especially in the background of COVI-19 pandemic. The manuscript is well written and the results are represented in a sufficient and clear way.

Response: We appreciate these favorable comments.

  1. In summary the manuscript can be recommended for publication, However, it would be desirable if the authors could add a paragraph describing how they would address the challenges mentioned here in further programs.

Response: Thank you for noting this. We have added in the discussion section: “Given the altered formats and timelines of training and the possibility of long-term adoption of these changes [4], future programs may attract different types of learners with different and varied motivations, expectations, and outcomes [21]. The development of new “hybrid” models using proven virtual components but with built-in, in-person activities better suited to discussion, idea generation, and team and relationship building may provide the best approaches for global health training and education in the post COVID-19 era [4].”

We have also written in the conclusions section: “Trainees perceived great benefit of program participation and mentorship. Moreover, integrating trainings regarding the mentor-mentee relationship for both trainees and faculty may provide the foundation for more successful interactions. However, trainees and faculty perceived communication and lack of clear expectations to be challenges, both of which may be compounded by the loss of in-person meetings and the ability to build strong relationships and shared understandings of expectations and roles, particularly relevant in collaborations across different cultural norms. Thus, it is essential to identify innovative ways to make virtual meetings and platforms more conducive to communication (e.g., project collaboration and planning tools) and relationship building (e.g., sharing personal information/photos, connecting via ice-breaker questions). This is particularly crucial given that many global health training programs were already shifting online, even before the COVID-19 pandemic. The likely long-lasting impact of the pandemic has solidified the need to ensure that global health research training programs capitalize on both in-person and virtual opportunities to enhance mentee-mentor relationships, facilitate better communication, and foster the achievement of trainee goals.”

Reviewer 3 Report

Dear author,

The manuscript title is “Research capacity training on environmental health and noncommunicable diseases in the country of Georgia: challenges and lessons learned during the COVID-19 pandemic and it aims to evaluate the experience of the participants on a training program on research capacity on environmental health and noncommunicable diseases developed in Georgia. As the program was launched in the early beginning of the pandemic, challenges that had arisen were also identified.

The topic falls within the aims and scope of the journal. It is relevant in this field as promoting mentoring programs and health communication is an urgent need. Plus, sometimes these programs are implemented but are not evaluated which means lost of precious information. It is very curious to verify the increasing importance of soft skills.

Some particular suggestions/comments will be done here:

-      Lines 30/31 – I would suggest not to repeat words in keywords that are already in the title

-      Figure 1 has not a good quality and you should add somewhere the meaning of “RCR” and “TBD”; delete the first legend of the Figure

-      Line 208 – how do you explain that not all faculty have answered?

-      Table 1 – footnote – I do not understand the last sentence as I do not see “1” and “3” anywhere

-      Table 2 – if it is going to stay in three different pages, please add again the legend on the second and third pages

-      Discussion – with only 15 participants it is a pity that you do not have a better qualitative analysis, why didn’t you did it? It would be the only improvement I would suggest. The discussion seemed poor also. Aren’t there other studies like yours to compare the expectations and lessons learned?

Author Response

We would like to thank the reviewers for their thoughtful feedback. Below we respond to each point. We appreciate the opportunity to enhance the quality of this manuscript by doing so.

Reviewer 3

  1. The manuscript title is “Research capacity training on environmental health and noncommunicablediseases in the country of Georgia: challenges and lessons learned during the COVID-19 pandemic” and it aims to evaluate the experience of the participants on a training program on research capacity on environmental health and noncommunicable diseases developed in Georgia. As the program was launched in the early beginning of the pandemic, challenges that had arisen were also identified. The topic falls within the aims and scope of the journal. It is relevant in this field as promoting mentoring programs and health communication is an urgent need. Plus, sometimes these programs are implemented but are not evaluated which means lost of precious information. It is very curious to verify the increasing importance of soft skills.

Response: Thank you for providing this favorable feedback.

Some particular suggestions/comments will be done here:

  1. Lines 30/31 – I would suggest not to repeat words in keywords that are already in the title

Response: We certainly agree that this seems redundant, but we also want to ensure that our keywords reflect the scope of the paper (irrespective of the title). We hope this reviewer finds this acceptable.

  1. Figure 1 has not a good quality and you should add somewhere the meaning of “RCR” and “TBD”; delete the first legend of the Figure

Response: Thank you for noting this. We have enhanced the quality of the figure, removed the unnecessary legend, and updated the figure accordingly.

  1. Line 208 – how do you explain that not all faculty have answered?

Response: Thank you for noting this. We were actually quite pleased with the response rate among faculty, as this was not a mandatory survey. The fact that all but 2 faculty participated we thought was a positive sign. However, we understand the reviewer’s point – that perhaps there were reasons faculty chose not to participate. Note, however, that the 2 faculty who did not participate were new to the program and thus, may not have felt that they knew enough to contribute to the evaluation of the program.

  1. Table 1 – footnote – I do not understand the last sentence as I do not see “1” and “3” anywhere

Response: Thank you for noting this. Reviewer 1 also raised concern about the potential lack of clarity of the footnotes in Table 1; thus, they were removed and instead integrated as appropriate into the text of the measures section and results section.

  1. Table 2 – if it is going to stay in three different pages, please add again the legend on the second and third pages

Response: Thank you for this suggestion. We have done so.

  1. Discussion – with only 15 participants it is a pity that you do not have a better qualitative analysis, why didn’t you did it? It would be the only improvement I would suggest. The discussion seemed poor also. Aren’t there other studies like yours to compare the expectations and lessons learned?

Response: We appreciate this comment. First, note that one of our primary objectives was to ensure high response rates so that we would have a representative sample of trainees and faculty; thus, we made the conscious decision to keep the assessment as brief as possible. Otherwise, we could have included a more rigorous qualitative assessment protocol. Unfortunately, online survey evaluations with open-ended questions are not ideal for comprehensive qualitative analysis in many cases, particularly when answers are sometimes quite brief (limiting the utility of comprehensive qualitative assessment protocols). For these reasons, we had to come to terms with the fact that some meaningful data could be yielded from the open-ended assessments, but that the respondent burden of a more rigorous protocol would have undermined the goal of the evaluation. As we begin to graduate trainees from the program, however, we plan to conduct exit interviews, which will provide the type of rich data this reviewer is alluding to.

            In addition, we appreciate this comment about the discussion section. The most novel component of this study is its timing – the evaluation of CARE captures this pivotal period coinciding with the COVID-19 pandemic. We were also surprised that there were limited publications regarding this. The literature that does exist was integrated into our discussion section. Nonetheless, we also place current findings into the context of the larger body of literature regarding mentorship, global health training programs, etc. The introduction section provides this type of context as well, so we wanted to ensure that the discussion section was not redundant but rather focused on the novel components of the current findings. We hope this reviewer appreciates this perspective.

Round 2

Reviewer 1 Report

The authors clearly improved the manuscript, therefore, in my opinion, it is now susceptible for publication